# Endocrine Regulation of Microvascular Receptor—Mediated Transcytosis and Its Therapeutic Opportunities: Insights by PCSK9—Mediated Regulation

**DOI:** 10.3390/pharmaceutics15041268

**Published:** 2023-04-18

**Authors:** Alexander D. Mazura, Claus U. Pietrzik

**Affiliations:** Institute of Pathobiochemistry, University Medical Center of the Johannes Gutenberg, University Mainz, Duesbergweg 6, 55128 Mainz, Germany

**Keywords:** blood brain barrier, receptor-mediated transcytosis, low-density lipoprotein receptor family, low-density lipoprotein receptor-related protein 1, proprotein convertase subtilisin/kexin type 9, central nervous system drug delivery, therapeutic blood–brain barrier modification

## Abstract

Currently, many neurological disorders lack effective treatment options due to biological barriers that effectively separate the central nervous system (CNS) from the periphery. CNS homeostasis is maintained by a highly selective exchange of molecules, with tightly controlled ligand-specific transport systems at the blood–brain barrier (BBB) playing a key role. Exploiting or modifying these endogenous transport systems could provide a valuable tool for targeting insufficient drug delivery into the CNS or pathological changes in the microvasculature. However, little is known about how BBB transcytosis is continuously regulated to respond to temporal or chronic changes in the environment. The aim of this mini-review is to draw attention to the sensitivity of the BBB to circulating molecules derived from peripheral tissues, which may indicate a fundamental endocrine-operating regulatory system of receptor-mediated transcytosis at the BBB. We present our thoughts in the context of the recent observation that low-density lipoprotein receptor-related protein 1 (LRP1)-mediated clearance of brain amyloid-β (Aβ) across the BBB is negatively regulated by peripheral proprotein convertase subtilisin/kexin type 9 (PCSK9). We hope that our conclusions will inspire future investigations of the BBB as dynamic communication interface between the CNS and periphery, whose peripheral regulatory mechanisms could be easily exploited for therapeutic purposes.

## 1. Introduction

Many disorders of the central nervous system (CNS) are lacking effective therapeutic treatment options due to the insufficiency of conventional drugs to enter the brain following peripheral administration [1]. Despite this fact, peripheral drug administration, in contrast to transcranial approaches that deliver drugs directly into certain brain segments [1], is still the application route of choice when it comes to handling, cost, and safety.

The majority of the extravascular CNS is separated from the providing vasculature by the blood brain barrier (BBB) and the blood–cerebrospinal fluid barrier (BCSFB) [2]. The BBB is formed of specific endothelial cells lining the inside of brain microvasculature vessels [3], while the BCSFB is established by the epithelial layer of the choroid plexus of each ventricle [4]. Although both are physical obstacles, the morphological and structural differences determine the central barrier functions, which are considered to be fundamentally different; whereas the BCSFB controls the composition of the cerebrospinal fluid (CSF), which is constantly produced by plasma ultrafiltration at a high turnover rate [4], the BBB ensures a highly selective exchange of ions, molecules, and immune cells between the compartments while maintaining a potent protection against detrimental blood-borne substances, toxins, or pathogens [3]. The heavily restrictive character of the BBB is due to a variety of proteinaceous junctions—in particular tight junctions—connecting adjacent brain microvascular endothelial cells to a close-mesh network of cells, which eliminates paracellular flow [5], has a very low pinocytosis rate compared to other endothelial cells of the vascular system [6], and has a presence of multiple efflux transporters within the endothelial plasma membrane, capable of rapidly excluding endogenous and drug metabolites that diffuse into the brain microvascular endothelium [7]. This clear distinction of the separate compartments at the brain microvasculature level is the foundation for the blood- (luminal) and brain-faced (abluminal) endothelial membrane terminology and their significantly different composition [8]. The vascular basement membrane surrounds the abluminal surface of the brain microvascular tubes and embeds pericytes in a comparatively high pericyte-to-endothelial cell ratio [9,10]. This is followed by the tight envelopment of the basal lamina by end-feet of perivascular astrocytes, providing a cellular link between the microvasculature and neuronal circuitry [11]. Following increasing evidence that the endothelial monolayer is in close exchange with these perivascular cells as well as proximal neurons and microglia, the concept of the neurovascular unit has been proposed, emphasizing the importance of extra-endothelial cells in the CNS in the maintenance and regulation of BBB integrity and function in a paracrine manner [12].

Since the extremely dense brain microcapillary system represents the largest blood–brain interface, regulating the immediate microenvironment of cerebral cells throughout the brain, and possesses the structural prerequisites for controlled but efficient transport, crossing the BBB is the primary, and in certain circumstances the exclusive route of entry for peripheral substances [13]. However, a fully functional BBB excludes nearly 100% of available large and more than 98% of small molecule drugs due to the size limitations (<0.4 kDa) and lack of high lipid solubility (<8 hydrogen bonds) required for non-specific diffusion across the BBB [1]. An alternative non-invasive approach to bypass the highly selective BBB features intranasal drug administration, where the therapeutic agent can be delivered across the nasal mucosa along the olfactory or trigeminal nerve pathways directly into the interstitial fluid of the CNS, but may also end up in the olfactory bulbs, vascular system, lymph nodes, or CSF, depending on factors such as substance properties, formulation characteristics, and handling [14]. Although promising for some therapeutic agents, several clinical trials have produced disappointing data [15,16,17,18], indicating that further optimization is needed to increase the brain parenchyma penetration and distribution for a variety of therapeutics intended to use the nasal-brain delivery route [19,20]. In contrast to BBB circumvention, several approaches [21], including microbubble-assisted focused-ultrasound or the use of hyperosmotic adjuvants [22,23], induce the temporary opening or disruption of BBB integrity to various extents, providing non-specific BBB permeability for a wide range of drugs. However, compromising BBB integrity, even slightly, is accompanied by an unregulated influx of plasma constituents that affect the sensitive CNS milieu, usually associated, among others, with inflammation, immune responses, and neuronal damage [24]. Even though some approaches show promising preclinical and early clinical evidence [22,23,25,26], long-term randomized clinical trials will be required to demonstrate whether these approaches are applicable for human therapy in different pathological contexts beyond the treatment of severe brain tumours.

Due to the absence of paracellular or transcellular channels at the brain microvasculature, the required transfer of larger and/or polar solutes through the semi-permeable BBB is physiologically performed by ligand-specific endogenous transport systems such as carrier- or receptor-mediated transcytosis [27,28]. The ligand-binding carriers or receptors can be heterogeneously embedded in the luminal or abluminal endothelial plasma membrane, thereby controlling the direction of transcellular transport [29,30,31]. Carrier transport systems facilitate the transport of small molecules, especially nutrients, down their concentration gradient (generally directed from blood-to-brain) [32], whereas transmembrane receptors specifically recognize diverse extracellular macromolecules and initiate the transcytosis of the specific ligand to the opposite plasma membrane [27,28]. A prominent ligand-independent endogenous transport system for polycationic molecules is the process of adsorptive-mediated transcytosis, in which positively charged substances attach to negatively charged microdomains of the endothelial membrane, followed by vesicle formation, transendothelial trafficking, and exocytosis at the opposite cell surface (generally directed from blood to brain) [33,34].

Exploiting endogenous transcellular transport systems for drug delivery into the brain would ideally allow for extensive drug distribution within the CNS by utilizing the widely branched brain microcapillary system without interfering with the structure and function of the BBB [35,36]. Individual structural reengineering for efficient brain penetration is possible for certain substances, including small molecules mimicking endogenous nutrient structures or macromolecules fused to receptor-binding motifs [37,38], but structural modifications of the therapeutic agent must still conform in terms of metabolism, pharmacokinetics, and therapeutic functionality, which have to be assessed on a cost- and time-intense substance-by-substance basis. Several nanocarrier systems, including liposomes, polymersomes, and metal-based nanoparticles, permit a greater structural flexibility of therapeutics [39,40,41]. Depending on the nanoparticle type and cargo properties, therapeutics such as monoclonal antibodies, antisense drugs, or short interfering RNA, are entrapped by, integrated in, or bound to a nanocarrier shell, which provides multiple advantages including the introduction of unique features (sensitive to magnetic fields or specific pH values), improved stability, solubility, bioavailability, biocompatibility, and permeability towards physiological barriers [39,40,41]. For example, liposome formulations with cationic gemini amphiphiles, depending on their stereochemistry, promote efficient transport across an in vitro human BBB model compared to free tracer or tracer-loaded neutral liposomes, thought to be due to enhanced adsorptive-mediated transcytosis [42]. In another example, nanocarriers labelled with dual ligands targeting the glucose transporter GLUT1, which is highly expressed at the BBB, showed significantly increased carrier-mediated transport into the mouse brain compared to single-ligand labelled or untargeted nanocarriers [43].

However, a well-known procedure to achieve efficient BBB transcytosis is the conjugation of short peptides or monoclonal antibodies mimicking ligand epitopes (peptidomimetics) to the nanocarrier surface, which are recognized by the respective endogenous receptors (“Trojan horse” strategy) [44]. Since enriched protein levels available at the luminal surface as well as rapid and efficient blood-to-brain transcytosis rates across the BBB are important characteristics of a potential vehicle receptor, the common targets are non-microvascular-specific metabolic receptors for insulin [45], transferrin [46], insulin-like growth factors [47], leptin [48], or low-density lipoprotein (LDL) [49]. For example, fluorescent nanoparticles conjugated to transferrin or LDL exhibit enhanced receptor-mediated transcytosis rates across an in vitro murine BBB model compared to unconjugated nanoparticles [50]. In addition, the transcytosis efficiency of transferrin-modified nanoparticles can be further enhanced by the introduction of a short cell-penetrating peptides onto the nanocarrier surface, which reduces endosomal entrapment and subsequent degradation of the delivery vehicles during receptor-mediated transcytosis [51]. Although drug delivery systems paired with moieties for receptor-mediated transcytosis are constantly being improved, brain penetration efficiencies are insufficient so far due to the potential competition with endogenous ligands and receptor availability on off-target cells, but also due to the tight regulation of receptor activity, especially at the brain microvasculature [41,52].

## 2. Receptor-Mediated Transcytosis at the BBB: Regulation of LDL Receptors

Transcytosis across an intact brain endothelial microvasculature is strictly downregulated [27]. Despite its relevance, little is known about the cellular regulatory mechanisms [53,54,55], but it has become clear that brain microvascular endothelial cells are sensitive to signals derived from extra-endothelial cells, particularly well-demonstrated by pericytes, part of the neurovascular unit, that significantly regulate BBB transcytosis [56,57,58]. However, how transcytosis is constantly modified due to temporal or chronic changes in the brain homeostasis and the peripheral environment, especially during aging or pathophysiological conditions, still raises many questions [28,59]. The scaling of transporter levels at the endothelial surface in response to individual cellular and environmental demands might be a central element in the dynamic transcytosis regulation at the BBB.

Members of the LDL receptor family, including the eponym LDL receptor and the prominent low-density lipoprotein receptor-related protein 1 (LRP1), are essential cell surface receptors participating in diverse physiological processes [60]. Receptors of the LDL receptor family share a modular domain organization through the use of repetitive characteristic substructures, and are divided into a cytoplasmic and extracellular section of variable length, connected by a single-pass transmembrane domain [61]. The cytoplasmic tail can contain several recognition motifs for cellular adaptor or scaffold molecules, responsible for clathrin-mediated internalization, cellular trafficking, or cell signalling [60]. The extracellular part harbours at least one cluster of ligand-binding repeats (cysteine-rich complement-type repeats) and an epidermal growth factor (EGF) precursor homology domain, which is essential for the pH-sensitive conformational change following endocytosis [61]. Differences in the number, composition, and position of each substructure provide the basis for the functional diversity of LDL receptor family members [62]. Interactions with other cell surface proteins (co-receptors) further modulate the ligand profile and downstream activities of these receptors [60]. The ubiquitously expressed LDL receptor is well known to mediate the cellular internalization of lipoprotein particles by binding to the associated apolipoproteins (Apo)B100 and E [63,64], primarily ensuring cellular cholesterol supply and thereby controlling the content of LDL, the most abundant cholesterol-carrying lipoprotein, in extracellular fluids [65]. In contrast to the modest quantity of potential ligands and functions of the LDL receptor, LRP1 is a multifunctional receptor participating in numerous physiological processes besides lipoprotein metabolism [66,67], including blood coagulation [68,69,70], clearance of various proteases and protease inhibitors [71,72], and immune responses [73,74,75,76,77,78], but also plays an essential role in various pathophysiological conditions, such as Alzheimer’s Disease (AD) [79,80,81]), atherosclerosis [82,83,84]), or cancer [85,86,87] through its interaction with at least 12 cytosolic adaptor proteins and the binding and internalization of more than 75 ligands, including ApoE, that vary significantly in structure and function [88].

LRP1 is expressed in certain cell types, such as hepatocytes, fibroblasts, macrophages, smooth muscle cells, neurons, and astrocytes, making it present in most tissues of the body [64,89]. In contrast to the high expression in most cell types [89], LRP1 is found at comparatively low levels in the brain microvasculature and supposed to be located predominantly in the abluminal (and to smaller quantities in the luminal) endothelial membrane [90,91,92].

Due to the structural prerequisites, members of the LDL receptor family are inherently endocytic receptors, which bind to specific ligands from the extracellular space and translocate in complex with the ligand from the plasma membrane into an intracellular vesicle. However, when it comes to extremely thin (~200 nm) and polarized endothelial cells of the brain microvasculature [6,8,37], the process of receptor-mediated endocytosis can be extended to receptor-mediated transcytosis, where complex vesicular sorting mechanisms and exocytosis at the opposite plasma membrane translocate the ligand from one extracellular fluid to another. Whereas the possibility of LDL receptor-mediated transcytosis is discussed controversially, brain-to-blood and blood-to-brain BBB transcytosis mediated by LRP1 has been documented for several ligands [91,93,94,95,96]. The high transcytosis rate of LRP1 and the tightly regulated low brain microvascular protein levels [92], which are significantly reduced with age or in AD [97,98,99], might reflect the significance and performance of LRP1 transcytosis at the BBB.

The receptor surface levels of members of the LDL receptor family can be regulated at different cellular levels in response to physiological or pathophysiological conditions. A central physiological feedback loop is transcriptional regulation due to cellular levels of a ligand. The cellular depletion of cholesterol initiates the two-step proteolysis of the transcription factor sterol-regulatory element binding protein (SREBP)-2 in the Golgi apparatus, which allows its translocation from the endoplasmic reticulum (ER) membrane into the nucleus [100]. While nuclear SREBP-2 enhances the transcription of the LDL receptor, it inhibits LRP1 transcription [101]. As cholesterol levels rise in the ER membrane, SREBP-2 translocation is prohibited, thereby returning the effect on the receptor transcription [100].

In addition to transcriptional regulation, cell surface activity of transmembrane receptors is limited by proteolytic shedding. Membrane-anchored metalloproteases, including ADAM10, ADAM17, and BACE1 are capable of cleaving adjacent LRP1 within the extracellular chain, releasing a large fragment into the extracellular space [102,103]. In contrast to secreted soluble LRP1, the remaining truncated receptor is unable to bind to extracellular ligands, which limits cell surface LRP1 activity.

Over the past two decades [104], an entirely different regulatory mechanism has moved into the centre of attention: the proprotein convertase subtilisin/kexin type 9 (PCSK9)-mediated degradation of several LDL receptor family members [105,106,107]. The serine protease PCSK9, a member of the proteinase K subfamily of subtilases, is expressed as a soluble zymogen primarily in the liver and to a lesser extent in the kidney, small intestine, and the brain [104], but not in brain microvascular endothelial cells [108]. Following the signal peptide (amino acids (aa) 1–30), pro-PCSK9 exhibits an N-terminal prodomain (aa 31–152) that precedes the catalytic subunit (aa 153–404), which contains the classical serine protease catalytic triad (Asp186, His226, Ser386) and the oxyanion hole (Asn317) [109,110,111,112]. The catalytic subunit is connected by a hinge region (aa 422–452) to a C-terminal Cys/His-rich domain (CHRD) (aa 453–692), which is composed of three tandem repeats (M1: aa 453–529, M2: aa 530–603, M3: aa 604–692) [109,110,111,112]. Maturation of pro-PCSK9 is achieved by Ca^2+^-independent autocatalysis of its prodomain early in the ER, which is required for its secretion into the extracellular space [104,113,114]. The separated prodomain remains non-covalently attached to the subtilisin-like catalytic site, preventing further enzymatic activity [104,109,110,111]. The mode of extracellular PCSK9 regulation, which is independent of its proteolytic activity, is well studied for the LDL receptor (Figure 1). Usually, cell surface LDL receptors bind to extracellular cholesterol-rich lipoprotein particles and internalize their ligands through clathrin-coated pits [65]. The LDL receptor/ligand complex enters the endosomal/lysosomal degradation pathway and dissociates pH-dependently (pH < 6) in sorting endosomes due to an EGF homology domain-induced conformational change in the LDL receptor, disrupting the lipoprotein binding sites [115,116,117]. While the released LDL receptor is recycled back to the cell surface, restoring the initial receptor level, the lipoprotein particle remains in the degradation pathway until lysosomal disintegration, releasing the transported cholesterol [65]. Analogous to receptor-mediated ligand endocytosis, extracellular PCSK9 is able to bind with its catalytic subunit to the EGF-A repeat within the EGF homology domain of the LDL receptor in a Ca^2+^-dependent manner, and enters the endosomal/lysosomal system in association with the receptor [118,119]. However, instead of complex dissociation, PCSK9-LDL receptor binding is enhanced with decreasing pH value, which, through an as-yet-unidentified mechanism(s), redirects the LDL receptor from the endosome to the lysosome for degradation rather than allowing recycling back to the cell surface [118,120]. Consequently, the LDL receptor level at the cell surface is reduced at the expense of extracellular PCSK9 molecules. Although not required for direct binding, the availability of the PCSK9 CHRD domain (aa 425–692), as well as at least three ligand-binding repeats and the β-propeller domain of the LDL receptor, are essential for subsequent LDL receptor degradation, indicating a more complex mechanism [120,121].

The PCSK9-mediated regulatory pathway extends the cholesterol-responsive negative feedback loop within cellular cholesterol homeostasis to the post-transcriptional level. In response to low cholesterol levels, PCSK9 and the LDL receptor are transcriptionally activated by SREBP-2 binding to the sterol-regulatory element site of the promotor sequences [122,123], are produced side by side as precursor proteins, and promote the efficient processing of each other [124]. Maturation of pro-PCSK9 via autocatalytic intramolecular cleavage in the ER is promoted by the binding to the EGF-A repeat of the pro-LDL receptor, whereas processed PCSK9 subsequently acts as an LDL receptor chaperone, supporting the transport of pro-LDL receptor molecules from the ER to the Golgi apparatus and towards the cell membrane after completion of glycosylation [124,125].

Although intracellular PCSK9 binding might occur, directing the bound LDL receptor directly from the trans-Golgi network to the lysosomes for degradation [121,126,127], a significant percentage of mature LDL receptor and PCSK9 proteins have to be integrated into the plasma membrane and secreted into the extracellular space, respectively, due to the principle of a delayed negative feedback loop. Subsequently, the majority of extracellular PCSK9 binds to target receptors in the immediate environment and is degraded, with only a small fraction entering the vascular system [126,128]. Plasma PCSK9, which is supposed to derive exclusively from the liver [128], is comparatively low in concentration (~0.05 µg/mL–~0.6 µg/mL) [105], which might indicate that circulating PCSK9, due to its self-destructive mode of action, primarily impacts cells and tissues with moderate to low target receptor levels and exerts a broader spectrum of regulatory functions than just cholesterol homeostasis.

## 3. Endocrine Regulation by PCSK9 and Its Inhibition

PCSK9 regulation might occur intracellularly, but especially extracellularly in an autocrine, paracrine, and endocrine fashion. In addition to cellular self-regulation of mature receptor concentrations at the cell surface, surrounding cells are capable of influencing the cell surface levels of PCSK9-regulated LDL receptor family members. There is evidence that neurons and human vascular smooth muscle cells secrete PCSK9, which might affect the abluminal side of the BBB [104,108]. Cerebral PCSK9 levels are considered to be independent of peripheral levels because PCSK9 does not cross an intact BBB [129]. Unlike for peripheral PCSK9, elevated levels of PCSK9 in brain fluids correlate with neurodegenerative disorders, including AD [130,131]. However, targeting cerebral PCSK9 levels might be as difficult as treating neurological symptoms and pathological mechanisms in the brain.

A more accessible PCSK9 reservoir is represented by liver-derived PCSK9 in the vascular system, which regulates cell surfaces in an endocrine manner. In a previous study, we identified circulating PCSK9 as a regulator of LRP1-mediated amyloid-β (Aβ) transport across the BBB (Figure 2) [132]. The development of the progressive neurodegenerative disorder AD is typically characterized by the increased levels of Aβ peptides [133,134], which accumulate extracellularly within the brain due to impaired brain clearance [135,136]. A central clearance pathway for brain Aβ is the continuous brain-to-blood transport across the BBB, which is highly dependent on cell surface LRP1 activity [90,95,137]. Microvascular LRP1 is supposed to be primarily located at the abluminal side and to a lesser extent at the luminal side of endothelial cells [90,91,92]. Using an established BBB in vitro model, we observed significantly reduced LRP1-mediated Aβ transport across a murine brain endothelial monolayer upon incubation with recombinant PCSK9 [132]. Consistently, we reversed this outcome by repetitively injecting FDA-approved monoclonal anti-PCSK9 antibodies into the peritoneum of an AD mouse model, resulting in substantially decreased cerebral Aβ concentrations and improved hippocampus-dependent learning behaviour compared to control-treated mice [132]. As peripheral PCSK9 inhibition was unable to reproduce these effects in brain endothelium-specific LRP1^−/−^ AD mice, our data may reveal the potential to modify BBB receptor quantities for therapeutic contexts [132]. Since LRP1 is lowly concentrated in the brain microvasculature and PCSK9 expression has not been detected in brain microvascular endothelial cells, externally derived PCSK9 in a paracrine and endocrine fashion might play an important role in the tight control of LRP1 at the BBB. This observation is not exclusive to LRP1 but includes all PCSK9-regulated receptors.

The use of the receptor-binding epitopes of ApoB and E allows endocytosis by members of the LDL receptor family and subsequent transcellular transport from the luminal to the abluminal membrane of brain endothelial cells. Subsequently, Apo-like structures are exocytosed into the CNS and internalized by proximal neurons or astrocytes [138]. Studies fusing the binding epitope of ApoB or E to recombinant proteins [138], enzymes [138,139,140], or nanoparticles [141,142,143], displayed promising brain penetration efficiencies. Unfortunately, and in line with similar studies targeting other metabolic receptors for ligands such as transferrin or insulin, non-cerebral tissues, particularly the liver, also showed elevated concentrations of the Apo-like structures due to ubiquitous receptor expression. Therefore, strategies to increase brain penetration and minimize off-target delivery to peripheral tissues are required to effectively treat CNS disorders.

Our results demonstrate that receptor-mediated Aβ clearance across the BBB by LRP1 can be significantly enhanced due to peripheral PCSK9 inhibition. Besides the use of high-affinity monoclonal antibodies at substantial concentrations in therapeutic contexts to silence circulating PCSK9, PCSK9 receptor affinity is physiologically reduced by furin cleavage or the binding to Apo-containing lipoproteins as demonstrated for LDL particles [144]. In addition, total peripheral target receptor concentration directly influences the plasma PCSK9 concentration because PCSK9 is degraded at the end of the regulatory pathway. It is therefore conceivable that some of these mechanisms might affect circulating PCSK9 more effectively than cellular, or just secreted, PCSK9, suggesting the possibility that brain microvascular receptor levels could be increased not only generally but also selectively without favouring receptors of off-target peripheral tissues.

## 4. Limitations, Potential Risks, and Future Directions

Illuminating the complex crosstalk between peripheral tissues, which strongly influence the plasma composition, and the CNS at the BBB as a dynamic endocrine interface, might reveal potential approaches to increase the efficiency of drug penetration via endogenous pathways.

PCSK9 is a potent post-transcriptional regulator of members of the LDL receptor family. Circulating, liver-derived PCSK9 reduces cell surface receptor concentration within the vascular system, including the brain microvascular endothelium, whereas inhibition of peripheral PCSK9 has the opposite effect. Therapeutic FDA-approved inhibition of circulating PCSK9 by peripheral application of monoclonal antibodies to raise cell surface LDL receptor levels is a more recently developed but well-established treatment for patients with high plasma LDL cholesterol [145]. PCSK9 inhibitors are typically used in patients at very high risk of atherosclerotic cardiovascular disease events suffering from severe hypercholesterolemia, diabetes mellitus, or advanced chronic kidney disease, and despite these circumstances, exhibit an overall well-tolerated safety profile [146]. However, the consequences of PCSK9 inhibition over long-term treatment periods and in different pathophysiological contexts are still elusive. Beyond cholesterol metabolism, PCSK9 has been implicated in cancer cell immunity and the promotion of vascular inflammation [147,148,149,150], illustrating that the explicit role of PCSK9 in physiological and pathophysiological processes remains to be further explored. In addition to the observation that endocrine regulatory mechanisms via the vascular system could be relatively poorly understood, therapeutic interventions targeting circulating PCSK9 might inadvertently affect multiple physiological and pathophysiological processes throughout the body beyond the microvasculature. It is therefore all the more important to gain a deeper understanding of how a systemic regulatory mechanism might act site-specifically. One possibility could be the cell type-specific combination of co-receptors or adaptor molecules, that specify the interaction pattern, with extracellular ligands of a ubiquitously expressed cell surface receptor. Thus, a cell could be sensitized to a systemic regulatory mechanism, and vice versa, by the sufficient expression of a set of proteins. Further investigation is needed to clarify the potential implications of PCSK9 and its inhibition and to elucidate whether endocrine-regulating PCSK9 can be specifically targeted to increase receptor-mediated transcytosis across the BBB without favouring off-target internalization by PCSK9-producing peripheral organs. These considerations are not limited to PCSK9 and the LDL receptor family as metabolic receptors, including insulin and transferrin receptors, at the brain microvascular endothelium might also be sensitive to circulating regulatory proteins.

The mere knowledge that receptors at the BBB might be sensitive to, and their number regulated by, vascular circulating molecules could be beneficial in several ways. Transient upregulation could improve the translocation of peripherally administered therapeutic agents, such as nanoparticles, into the brain. In contrast, receptor downregulation could decrease off-target brain penetration and subsequent side effects of highly concentrated circulating drugs (e.g., t-Pa). As shown for enhanced LRP1-mediated Aβ clearance due to peripheral PCSK9 inhibition, altering the cell surface receptor concentration itself could be a therapeutic option by increasing or decreasing receptor-mediated transcytosis to or from the CNS. This would avoid the need to introduce a therapeutic agent into the CNS, and potential cross-reactions. BBB permeability changes dynamically due to aging or pathophysiological conditions, such as AD, which could be addressed by exploiting regulatory mechanisms as a preventive measure or therapeutic option after disease onset. Due to the relative simplicity of access and monitoring, endocrine regulatory pathways as described for circulating PCSK9 would be predestined. Thus, a deeper understanding of how BBB transport systems are organized and regulated might be crucial to efficiently exploit the BBB transcytosis machinery for drug delivery to the brain, and to modify the compromised BBB during pathophysiological states as a therapeutic option.

## Figures and Tables

**Figure 1 pharmaceutics-15-01268-f001:**
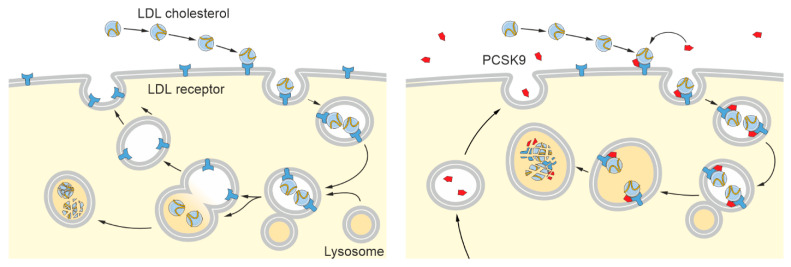
Mechanism of extracellular PCSK9 regulation to downregulate cell surface receptor levels. Left panel: Members of the low-density lipoprotein (LDL) receptor family bind to ligands (e.g., LDL receptors to LDL cholesterol) in the extracellular space. Clathrin-mediated endocytosis translocates the receptor/ligand complex into endosomes, where acidification causes the receptor and ligand to dissociate. The free receptor can be recycled back to the cell surface while the ligand remains in the endosomal/lysosomal degradation pathway. Right panel: Extracellular proprotein convertase subtilisin/kexin type 9 (PCSK9) is capable of binding to several members of the LDL receptor family, including its eponym, at the cell surface. After internalization, the binding strength of PCSK9 to the receptor increases at a slightly acidic pH, trapping the receptor in the intracellular compartment and leading to its lysosomal degradation and reduced cell surface receptor levels.

**Figure 2 pharmaceutics-15-01268-f002:**
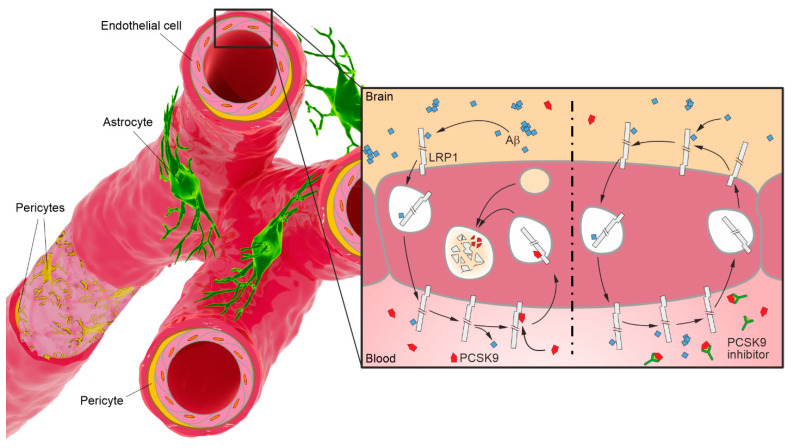
Simplified model of the PCSK9-regulated Aβ transcytosis across the BBB, mediated by LRP1, and its inhibition via monoclonal antibodies. Left panel: Low-density lipoprotein receptor-related protein 1 (LRP1) binds to cerebral amyloid-β (Aβ) and initiates its translocation into the vascular system across the blood–brain barrier (BBB) along its concentration gradient. Secreted PCSK9 in the brain and periphery binds to LRP1 at the endothelial cell surface and targets the receptor for lysosomal degradation, reducing the amount of available Aβ clearance receptors in the microvasculature. Right panel: The peripheral application of monoclonal antibodies targeting circulating PCSK9 diminishes the number of PCSK9 molecules capable of binding to LRP1, which increases the amount of LRP1 recycled back to the cell surface and consequently the rate of Aβ transcytosis from brain to blood.

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
