# Peer review of "Endocrine Regulation of Microvascular Receptor—Mediated Transcytosis and Its Therapeutic Opportunities: Insights by PCSK9—Mediated Regulation"

_pharmaceutics, 2023, doi:10.3390/pharmaceutics15041268_

Round 1

Reviewer 1 Report

The review discusses the potential for targeting endocrine regulatory pathways to increase the efficiency of drug penetration through the blood-brain barrier (BBB) and into the central nervous system (CNS). Specifically, the review mentions PCSK9 as a regulator of LDL receptors in the vascular system, including the brain microvascular endothelium, and suggests that inhibiting peripheral PCSK9 could increase receptor-mediated transcytosis across the BBB without off-target effects.

Although this review is interesting it needs to be improved to be accepted for publication:

1. Clarifying the limitations and potential risks of the approach: Any scientific work should clearly acknowledge the limitations and potential risks of the approach being proposed. This would help to ensure that researchers and practitioners are aware of the potential issues and can take steps to mitigate them.

2. the author need to prepare figures for the following sections:

a. 2.REZEPTOR-MEDIATED TRANSCYTOSIS AT THE BBB: REGULATION OF LOW- 1 DENSITY LIPOPROTEIN RECEPTORS BY PCSK9

b. ENDOCRINE PCSK9 REGULATION AND ITS INHIBITION

since the review is all text it will not be very appealing to the readers and and also use of graphics will make is easier to understand the concept. 

The authors should also consider making a graphic abstract.

Author Response

Point to point response is attached bellow

Reviewer 2 Report

I believe that the review submitted for publication is useful to a wide range of specialists working in the field of neurodegenerative diseases.  I have no comments on the present text. I encourage the authors to publish new data from their research on the topic. 

Author Response

(The authors gave the same response as above.)

Reviewer 3 Report

The review " Endocrine regulation of microvascular receptor-mediated transcytosis and its therapeutic opportunities: insights by PCSK9-mediated regulation"  by Mazura and Pietrzik deals with the key role that endogenous transporter  might play in the delivery of pharmaceuticals to the central nervous system. In particular, the author focused the attention on the sensitivity of the BBB to circulating molecules derived from peripheral tissues, which seems to suggest an endocrine-operating regulatory system of receptor-mediated transcytosis at the BBB.The ultimate goal of the review is to highlight how the BBB should be viewed as a dynamic communication interface between the central nervous system and the periphery and how this communication can be exploited for drug delivery. With this goal in mind, the authors reported their reflections on the specific case of the effect of peripheral proprotein convertase subtilisin/kexin type 9 (PCSK9) on low-density lipoprotein receptor-related protein 1 (LRP1) mediated amyloid-β (Aβ) clearance through the BBB.

The review is clear and the authors' goal was completely achieved. However, in my opinion, the introduction needs revision because the recent literature (last 3-5 years) is missing. In fact, recent reviews on advances in drug delivery through the BBB and articles describing specific drug delivery systems should be added (e.g., B. Simonis, D. Vignone, O. Gonzalez Paz et al. Journal of Colloid and Interface Science 627 (2022) 283–298; Y. Zhou et al. Journal of Controlled Release 270 (2018) 290–303; Qian Lu et al ACS Appl. Bio Mater. 1 (2018) 1687−1694; S. Lakkadwala, J. Singh Colloids and Surfaces B: Biointerfaces 173 (2019) 27–35; Mária Mészáros et al European Journal of Pharmaceutical Sciences 123 (2018) 228-240; Denzil Furtado et al Adv. Mater. 30 (2018) 1801362)

Author Response

Pion to point response is attached bellow

Round 2

Reviewer 1 Report

I wish to commend the authors for addressing all my comments and making improvements to the revised manuscript.